# Prognostication of Poor Survival After Cardiac Resynchronization Therapy

**DOI:** 10.3390/medicina56010019

**Published:** 2020-01-04

**Authors:** Paulius Bašinskas, Neris Stoškutė, Austėja Gerulytė, Agnė Abramavičiūtė, Aras Puodžiukynas, Tomas Kazakevičius

**Affiliations:** 1Department of Cardiology, Medical Academy, Faculty of Medicine, Lithuanian University of Health Sciences, 50009 Kaunas, Lithuania; nerisstoskute@yahoo.com (N.S.); austeja.ger@gmail.com (A.G.); Abramaviciute.agne@gmail.com (A.A.); aras.puodziukynas@gmail.com (A.P.); kazakev@gmail.com (T.K.); 2Kaunas Region Society of Cardiology, Kaunas 50009, Lithuania

**Keywords:** heart failure, cardiac resynchronization therapy, biventricular pacing, ischemic cardiomyopathy, pulmonary hypertension, echocardiography

## Abstract

*Background and Objectives:* Cardiac resynchronization therapy (CRT) is a successful treatment option for appropriately selected patients. However, one–third of recipients do not experience any positive outcome or their condition even declines. We aimed to assess preimplantation factors associated with worse survival after the CRT. *Materials and Methods:* This was a retrospective unicenter trial. The study cohort included 183 consecutive CRT-treated patients. Baseline demographic, clinical, electrocardiographic, and echocardiographic characteristics were analyzed. *Results:* After the median follow-up of 15.6 months (9.3–26.3), 20 patients had died (11%). In multivariate Cox regression analysis, ischemic origin of heart failure (HF) was a significant predictor of poor survival (adjusted hazard ratio (aHR) 15.235, 95% confidence interval (CI) (1.999–116.1), *p* = 0.009). In univariate Cox regression, tricuspid annular plane systolic excursion (TAPSE) <15.5 mm (sensitivity 0.824, specificity 0.526; HR 5.019, 95% CI (1.436–17.539), *p* = 0.012), post-implantation prescribed antiplatelet agents (HR 2.569, 95% CI (1.060–6.226), *p* = 0.037), statins (HR 2.983, 95% CI (1.146–7.764), *p* = 0.025), and nitrates (HR 3.694, 95% CI (1.342–10.171), *p* = 0.011) appeared to be related with adverse outcome. *Conclusions:* ischemic etiology of HF is a significant factor associated with worse survival after the CRT. Decreased TAPSE is also related to poor survival.

## 1. Introduction

Heart failure (HF) is a major public health problem associated with significant hospital admission and mortality rates, as well as costly health care expenditures [1]. It is estimated that the average one-year mortality rate for stable (ambulatory) and hospitalized patients is 7% and 17%, respectively [2]. The main factors associated with HF prognosis are male gender; age; elevated resting heart rate; hospital admissions and emergency department visits the previous year; natriuretic peptides; troponin T and I levels; and comorbidities: diabetes mellitus, chronic kidney disease, dementia, cirrhosis, atrial fibrillation, cardiac conduction disorders, ischemic origin of HF, hypertensive heart disease, etc. [3,4,5].

Cardiac resynchronization therapy (CRT) is an established treatment for symptomatic New York Heart Association (NYHA) class III–IV patients resistant to treatment with optimal medical therapy, who have left bundle branch block (LBBB), QRS >130 ms, and left ventricular (LV) ejection fraction <35% [6]. CRT improves symptoms and quality of life and reduces mortality and morbidity [7,8]. Patients who undergo implantation of a cardiac resynchronization device with (CRT-D) or without a defibrillator (CRT-P) have, respectively, 37% and 34% lower risk of hospitalization or death compared with only optimal medical therapy [9]. For patients with LBBB and common indications for treatment with an implantable cardioverter-defibrillator, CRT-D is more beneficial than an implantable cardioverter-defibrillator alone [6]. On the other hand, patients treated with CRT-P and CRT-D have shown similar results, and neither is superior [10,11].

Despite the recent progress, 30%–40% of patients do not respond favorably to CRT, or their condition even declines [1,12]. Due to the high rate of non-response, in recent years, there have been numerous studies aiming to assess baseline factors associated with CRT outcome. Atrial fibrillation (AF) is often related to poor prognosis, but the results are incomplete and their analysis is complicated by the lower rate of biventricular stimulation, which itself is a negative prognostic marker [13,14,15,16,17,18]. QRS length more than 150 ms and predisposed by LBBB is usually considered a class Ia indication for CRT, but it is still under dispute whether LBBB or QRS duration is a predominant factor [13,19]. 

Ischemic cardiomyopathy is another issue. The Predictors of Response to CRT (PROSPECT) trial concluded that the nonischemic etiology was associated with more significant improvement in clinical state and LV reverse remodeling [20]. According to The Cardiac Resynchronization—Heart Failure (CARE-HF) study, the ischemic origin of HF is related to a lower rate of LV reverse remodeling but does not have an impact on mortality, hospitalizations, or changes in NYHA class [21,22,23]. Besides, patients with ischemic cardiomyopathy or AF are usually older and have more comorbidities. That leads to further worsening of their condition and impedes analysis of clinical trials.

Echocardiographic parameters to predict CRT response have also been widely investigated. Although studies have proposed some single prognostic factors or their combinations, in prospective multicenter double-blinded trials, neither was eligible to guide for CRT patient selection [24,25].

Our study aimed to assess the baseline prognostic factors related to worse survival after CRT.

## 2. Materials and Methods

### 2.1. Study Design

This was a retrospective study to review the patients who underwent implantation of a cardiac resynchronization device at the Hospital of Lithuanian University of Health Sciences between January 2014 and May 2019. Data were manually collected between 10 March and 15 May 2019, in the “Hospital Information System”. Participants were included according to The Australian Refined Diagnosis Related Groups (AR-DRGs) classification codes Z95 and I50. We included both patients who had novel device implantation and those who upgraded from previous right ventricular pacing. Missing echocardiographic measurements were calculated using EchoPac (GE Healthcare, Horten, Norway) software. Patients who lacked the below mentioned clinical data in their records or whose echocardiographic images were not available to assess were excluded. Equally, we omitted three patients who died during the first three months of the follow-up and those who later experienced cardiac resynchronization device removal without replacement. Finally, 183 cases proceeded to further analysis. We divided the total cohort into two sub-groups according to survival status, which was evaluated on 13 May 2019.

The study protocol was approved by Kaunas Regional Biomedical Research Ethics Committee at the Lithuanian University of Health Sciences (No. BE-2-86) and all subjects provided written informed consent.

### 2.2. Study Population Characteristics

The following demographic, clinical, electrocardiographic, and echocardiographic characteristics were analyzed:(a)Gender, age;(b)Type of the implanted device (CRT-D or CRT-P);(c)Echocardiographic characteristics: LV end-diastolic diameter and index (LVEDD, LVEDDi), LV systolic and diastolic volumes and indices (LVESV, LVESV(i), LVEDV, LVEDV(i)), LV ejection fraction (LVEF) and diastolic function, left atrium diameter (LA) and volume (LAV), right atrium (RA) and right ventricular (RV) diameters, tricuspid annular plane systolic excursion (TAPSE), pulmonary artery systolic pressure (PASP), and grade of mitral regurgitation. All measurements were performed following guidelines of the American Society of Echocardiography and the European Association of Cardiovascular Imaging [26,27,28]. A conventional echocardiography system, Vivid 7, GE-Vingmed Ultrasound AS, Horten, Norway with a 3.5 MHz transducer, was used. The echocardiographic analysis was made using EchoPac (GE Healthcare, Horten, Norway) software;(d)Functional status: 6 min walking test (6MWT), NYHA functional class;(e)Comorbidities: diabetes, chronic obstructive pulmonary disease, chronic kidney disease, malignancies, permanent AF, hypertensive heart disease, and ischemic cardiomyopathy. Ischemic etiology was considered when a patient had one or more of the followings: myocardial infarction, coronary artery bypass surgery, percutaneous transluminal coronary angioplasty, at least one coronary narrowing more than 75%, or left main artery stenosis more than 50%;(f)Medications: beta-adrenoceptor blockers (BB), angiotensin-converting enzyme (ACE) inhibitors, angiotensin II receptor blockers (ARBs), mineralocorticoid antagonists, loop diuretics, antiplatelets, anticoagulants, glycosides, antiarrhythmics, statins, and nitrates.

### 2.3. Statistical Analysis

Statistical analysis was performed with IBM SPSS 22.0 software. Categorical variables were presented as a percentage and were analyzed with chi-square statistics. The Shapiro–Wilk test was used to assess the normality of continuous data. Non-Gaussian distributed variables were described as median with 25–75th percentiles and analyzed with Mann–Whitney U test, normally distributed variables were expressed as mean ± standard deviation (SD) and compared using Student’s t test. To find out the significance of parameters in predicting the CRT outcome, receiver operating characteristic (ROC) curves were used. The optimal cut-off points were selected based on the maximal Youden index (sensitivity + specificity − 1). Cox regression survival analysis was applied to evaluate the predictive value of multiple factors on mortality. Variables that were significant in univariate analysis were adjusted for the multivariate analysis. To compare the risks of observed mortality, Cox proportional hazard models were used. Kaplan–Meier curves and the log-rank test were used to assess observed cumulative survival. A two-sided *p* value of 0.05 was considered statistically significant. 

## 3. Results

### 3.1. Study Population Characteristics

Over a study period of 63.6 months, a total of 183 patients were included. Most of the patients were senior men with the mean age of 66.4 ± 11.4 years. There were 155 (84.7%) novel implantations. Complete left bundle branch block was observed in 76.6% patients and wide QRS duration (>130 ms) in 84.8% of patients, while average QRS duration was 165.3 ± 31.8 ms. Two-thirds of the group had CRT-P and NYHA functional class III. Hypertensive heart disease occurred in 82.2%, atrial fibrillation 30.7%, and diabetes in 22.2% of the cases. Ischemic etiology of HF was more common than non-ischemic. Pathogenetic heart failure medical treatment was prescribed as follows: Angiotensin-converting enzyme (ACE) inhibitors or angiotensin II receptor blockers (ARB) were received by 69.7%, beta-adrenoceptor blockers by 81.9%, and mineralocorticoid antagonists by 66.7% of patients. Loop diuretics were taken by 68.4%, amiodarone by 20.3%, and oral anticoagulants by 52.0% of the patients (Table 1).

### 3.2. Survival Prognostication

During a median follow-up of 15.6 months (9.3–26.4), 20 (11%) patients died. The median follow-up duration of survivors was 39.3 months (23.7–57.1), of the dead, 15.2 months (7.4–26.7).

There were relatively more males in the fatal outcome group. The predominant cause of HF in this group was ischemic cardiomyopathy. Moreover, chronic kidney disease was more prevalent in the fatal outcome group. Indeed, patients of the survivors’ group demonstrated significantly better right ventricular function, as indicated by superior TAPSE and lower pulmonary hypertension. Patients of the fatal outcome group were more often prescribed with statins, antiplatelets, and nitrates (Table 1).

During the follow-up, a more considerable relative increase (Δ) in LVEF in the survivors was noted compared with that of the fatal outcome group (34.7% (0; 66.7) vs. 0 (−18.8; 60.0), *p* = 0.054). There were no differences between the groups in other analyzed parameters during the follow-up.

According to the ROC analysis, baseline TAPSE values lower than 15.5 mm and PASP values higher than 39.5 mmHg were associated with an increased risk of death (Table 2). 

In the Kaplan–Meier survival analysis, male gender, ischemic origin and TAPSE <15.5 mm were related to decreased survival. PASP values less than 39.5 mmHg were also linked to a better prognosis (log-rank = 0.107). The survival curves according to these predictors are shown in Figure 1, Figure 2 and Figure 3.

Gender; ischemic origin; TAPSE; PASP; and treatment with antiplatelets, statins, and nitrates were included in the univariate Cox regression analysis. Ischemic etiology and treatment with statins, nitrates, antiplatelets, and TAPSE were significantly associated with the decreased survival after the CRT. Multivariate Cox regression analysis was performed to acknowledge the independent effect of these predictors. Ischemic cardiomyopathy was established as a significant independent risk factor associated with worse survival after the CRT (Table 3).

## 4. Discussion

This was a retrospective single-center trial comprised of all patients who underwent cardiac resynchronization device implantation in our hospital from January 2014 to May 2019. In contrast to other trials, we included all patients independently of their heart rhythm status (sinus rhythm or AF), cause of the conduction disorder, origin of the HF, NYHA class, and their comorbidities. We included both novel implantations and upgrades from a permanent pacemaker or implantable cardioverter-defibrillator. Moreover, we could precisely evaluated patients’ survival as we had access to the national database.

The main limitation of this study is a retrospective single-center design with relatively small sample size. Therefore, we could not take into account all the risk factors, clinical situations, psychoemotional factors, and each of the newest concepts. The latest studies in this field also report significant findings about the importance of patients’ nutritional status and body mass index before the CRT and their impact on treatment outcomes [29]. Moreover, heart failure related hospitalizations during the follow-up were not assessed, and information about LV lead position was also not available. In the near future, we intend to extend our CRT research in a more detailed form and consider the limitations of this study.

Comparing the baseline of our study participants’ characteristics with patients enrolled in the CRT Survey II, we found that the median age of our population was below the median of the European survey (67 years (60–75) vs. 70 years (62–76). In both studies there was a similar rate of males (78% vs. 76), patients with LBBB (77% vs. 73%), atrial fibrillation (30.7% vs. 26.0%), QRS duration >130 ms (84% vs. 87%), and CRD-P implantations (69% vs. 70%) involved. Our study included more patients with NYHA class III–IV (76.1% vs. 60%) and novel device implantation (85% vs. 72%). On the other hand, our participants had more advanced systolic heart failure (LVEF 23% (18–30%) vs. 29% (23–34%)) and ischemic etiology was more often an underlying cause of heart failure (56.1% vs. 45%). There were slight differences in adherence to pathogenetic HF medications at discharge: Beta-blockers were prescribed for 81.9% of our patients and 89% of the CRT Survey II participants, ACE inhibitor or ARB blockers for 69.7% vs. 89%, and mineralocorticoid antagonist for 66.7% vs. 63%, respectively [30].

As we have seen, not all participants of our study totally matched conventional CRT indications (NYHA class ≥III, LBBB as a cause of conduction disorder, QRS duration more than 130 ms). The underlying reason is that patients do not have equally pronounced CRT indications, for example, there can be relatively asymptomatic (NYHA class <III) patients with prolonged QRS duration who require defibrillator implantation for the prevention of sudden cardiac death or, contrarily, patients with moderately reduced left ventricular ejection fraction with high NYHA class and QRS >130 ms. Until 2016, guidelines recommended CRT for patients with QRS 120–150 ms [13], that explains why 15.2% of our participants’ QRS duration was shorter than 130 ms. Finally, decision making in clinical practice is often more complicated than described in medical guidelines, and physicians must take into account multiple clinical factors.

After a univariate adjustment, male gender; application of statins, nitrates, and antiplatelet agents; the ischemic origin of heart failure; TAPSE <15.5 mm; and PASP ≥39.5 mmHg appeared to be indicators of adverse long term survival. A univariate association between these medications and the unfortunate outcome could be explained because they are used for the pathogenetic treatment of ischemic cardiomyopathy and its complications. As we have seen after a multivariate adjustment, the ischemic origin of heart failure remained the only significant prognostic marker associated with worse survival after the CRT. Statins, nitrates, nor antiplatelets negatively impacted on CRT outcome.

Researchers have not yet concluded whether the ischemic origin of heart failure is associated with worse survival after the CRT [6,13]. Patients with ischemic cardiomyopathy are usually older and have more chronic diseases and life-long risk factors. Both systolic and diastolic dysfunctions are more severe. Intrinsically, their prognosis is worse than patients whose origin of HF is not ischemic [31]. Cardiac Resynchronization-Heart Failure (CARE-HF) and REVERSE (Resynchronization reverses Remodeling in Systolic Left Ventricular Dysfunction) trials established that all patients irrespective of the origin of HF experience similar clinical outcomes in terms of mortality and heart failure related hospitalizations, though the magnitude of left ventricular reverse remodeling is less prominent for those with ischemic etiology. However, in the REVERSE trial, only patients with NYHA class I–II were included. Today, CRT is not indicated for this subgroup of patients except for special circumstances. Both trials did not enroll patients with conventional pacemaker or implantable cardioverter-defibrillator indications or those not in sinus rhythm. They included only novel implantations [19,32,33]. Therefore, these trials underrepresent the CRT population of today.

The most recent clinical trials led by Francisco [5] and Jian-Shu [34] found that ischemic cardiomyopathy is a significant predictor related to worse survival after the CRT. The underlying mechanism of this phenomenon could be explained by CRT correcting conduction disorders caused by delayed activation regions of the left ventricular wall. However, when the synchronicity of contraction is impaired by myocardial scar tissue, it becomes more challenging to achieve an adequate rate of biventricular stimulation and higher activation rates are required. The situation becomes even more complicated when the lead is placed in the scar tissue, which leads to further limitations of the CRT effect.

Reduced RV function has been associated with adverse prognosis in HF patients, but there is less data available about whether preimplantation RV function is associated with CRT outcome. Our study showed that decreased baseline RV function expressed by TAPSE is related to poor survival after the CRT. Values of TAPSE represent RV function and are also negatively affected by pulmonary hypertension, which in our study also appeared to be related to the adverse CRT outcome. Usually, TAPSE evaluation is part of a routine echocardiographic examination, so if TAPSE could lead to reliable CRT outcome predictions, it would be simple to use in clinical practice. However, the role of this and other RV function parameters in CRT patient selection is still under dispute. Trials demonstrate different TAPSE cut-off values associated with CRT outcome (15 mm [35], 17 mm [36]) but their diagnostic accuracy is only modest. In single studies there have also appeared other parameters, such as tricuspid lateral annular systolic velocity >9 cm/s and RV global longitudinal strain >12.45%, with very high sensitivity and specificity in predicting LV reverse remodeling [37] or even decreased survival (RV global longitudinal strain <10.04%) [38]), but meta-analysis by Sharma concluded that TAPSE, RV ejection fraction, RV basal strain, and RV fractional area change were not able to predict CRT response as assessed by change in LVEF [39]. In summary, due to the different definitions of CRT outcomes and the small number of appropriate trials, any practical recommendations can be admitted.

Pulmonary hypertension is another factor correlating with poor prognosis in patients with HF. However, there is a lack of data about whether it also affects the outcome of the CRT. In retrospective studies by Wang [40] and Stern [41], higher baseline values of PASP (PASP >45 mmHg and PASP >50 mmHg, respectively) were associated with higher rates of death, heart transplantations, and repeated hospitalizations but not with the decreased LV reverse remodeling. A prospective study by Chatterjee denied the impact of pulmonary hypertension on the CRT outcome [42]. In our study we found that PASP values >39.5 mmHg were related to increased mortality after the CRT, though diagnostic accuracy was only modest and results were not confirmed in the Cox regression analysis. This cut-off value is close to 38 mmHg, which is supposed to be an indicator of present pulmonary hypertension [43]. Scientific evidence is not conclusive, so in the case of suspicious pulmonary hypertension, more detailed clinical evaluation is required. Finally, taking into account our findings about the negative prognostic value of decreased right ventricular function and the presence of pulmonary hypertension, more detailed RV function examination including 3D echocardiographic RV ejection fraction evaluation and long axis function assessment are required.

In the context of larger clinical trials, our research has an additive prognostic value in CRT patient selection. Due to our study design, any strict practical recommendations can be admitted. However, these findings can act as a stimulating factor to further larger-scale prospective studies to ensure the real scientific impact of heart failure etiology and right ventricular function on CRT outcome.

## 5. Conclusions

This study has revealed that ischemic origin of heart failure is a significant predictor related to worse survival after CRT. Decreased right ventricular function is also a negative prognostic factor associated with inferior long term outcome. Patients require further clinical evaluation and reconsideration of CRT indications in case of ischemic cardiomyopathy, severe RV dysfunction, or pulmonary hypertension.

## Figures and Tables

**Figure 1 medicina-56-00019-f001:**
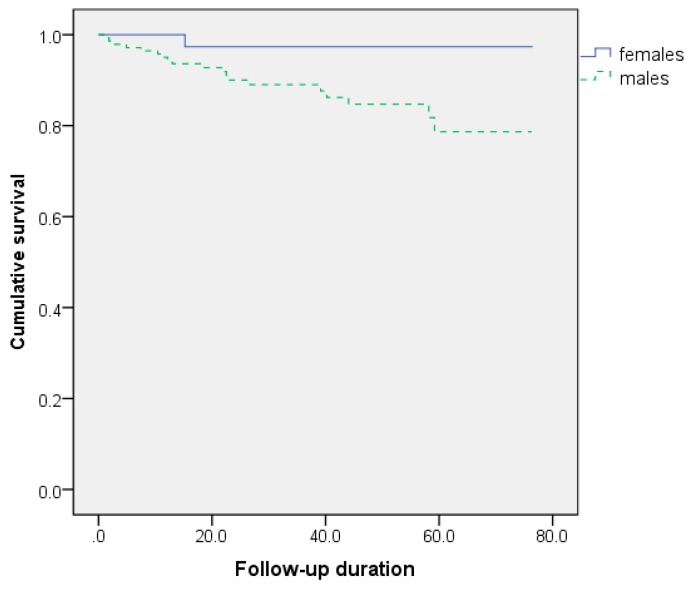
Gender influence on survival after cardiac resynchronization (log–rank *p* = 0.045).

**Figure 2 medicina-56-00019-f002:**
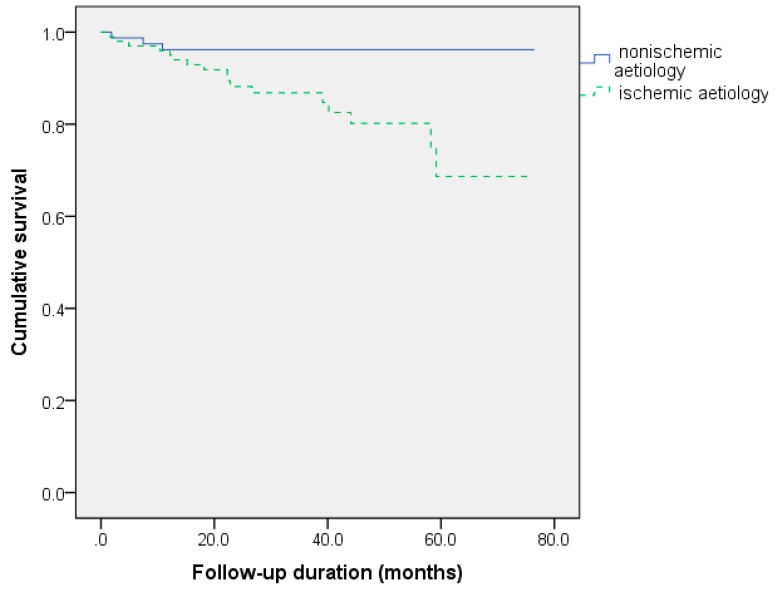
Survival differences after cardiac resynchronization depending on the origin of heart failure (log-rank *p* = 0.004).

**Figure 3 medicina-56-00019-f003:**
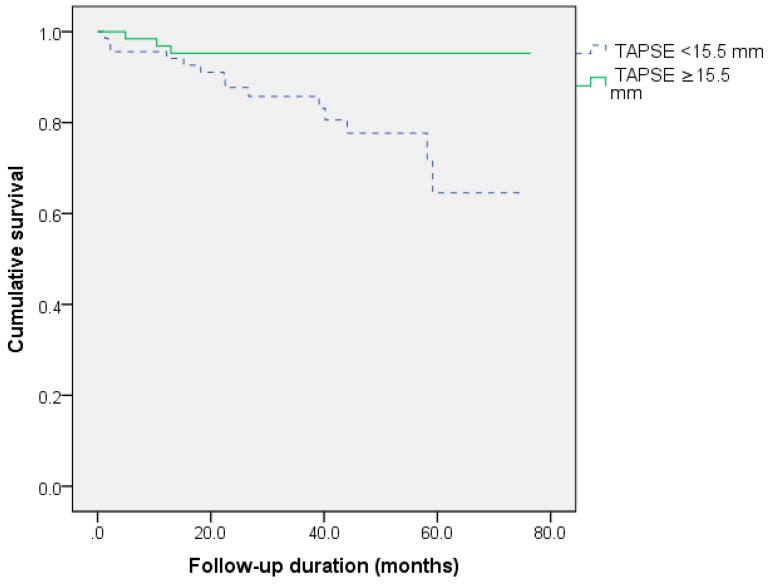
Impact of right ventricular function on survival after cardiac resynchronization (log-rank *p* = 0.005). TAPSE, tricuspid annular plane systolic excursion.

**Table 1 medicina-56-00019-t001:** Characteristics of the study population.

Characteristic	Total Population (N = 183)	Survivors (N = 163)	Fatal Outcome (N = 20)	*p*
Male gender, n (%)	142 (77.6)	123 (75.3)	19 (95.0)	0.047
CRT-P, n (%)	127 (69.4)	116 (71.0)	11 (55.0)	0.144
NYHA class ≥III, n (%)	139 (75.9)	124 (76.1)	15 (75.0)	0.911
Chronic kidney disease, n (%)	29 (16.1)	24 (14.5)	6 (30.0)	0.076
Ischemic cardiomyopathy, n (%)	103 (56.1)	85 (52.2)	17 (85.0)	0.005
**Echocardiographic Characteristics**				
LVEF (%) (Q_1_–Q_3_)	23.0 (18.0–30.0)	23.5 (18.0–30.0)	21.0 (17.3–25.8)	0.544
MR ≥II°, n (%)	102 (55.7)	89 (54.6)	13 (65.0)	0.425
LVEDD (mm) (Q_1_–Q_3_)	64 (57.0–69.0)	64.0 (57.0–69.0)	68.0 (56.0–71.0)	0.795
LVEDDi (mm) (Q_1_–Q_3_)	31.6 (27.9–35.4)	31.5 (27.9–35.4)	32.2 (27.1–36.5)	0.608
LVEDV (mL) (Q_1_–Q_3_)	176 (140.8–225.0)	174.0 (136.5–224.5)	181.0 (150.5–296.5)	0.552
LVEDVi (mL/m^2^) (Q_1_–Q_3_)	89.8 (68.6–113.3)	88.2 (68.1–112.1)	94.3 (76.4–169.6)	0.588
LVESV (mL) (Q_1_–Q_3_)	137.0 (100.0–182.5)	135.0 (98.0–180.0)	144.0 (113.5–228.0)	0.552
LVESVi (mL/m^2^) (Q_1_–Q_3_)	68.9 (50.4–93.7)	68.5 (49.2–91.6)	69.2 (58.8–114.9)	0.978
LA size (mm) (Q_1_–Q_3_)	50.0 (44.0–55.0)	49.0 (44.0–54.0)	50.0 (47.0–58.0)	0.145
LA volume (mL) (Q_1_–Q_3_)	103.0 (73.5–123.8)	101.5 (71.8–127.0)	104.0 (78.0–112.0)	0.606
LA volume index (mL/m^2^) (Q_1_–Q_3_)	49.2 (38.1–60.3)	49.9 (36.0–59.9)	48.4 (38.8–58.9)	0.596
RV size (mm) (Q_1_–Q_3_)	36.0 (33.0–41.5)	37.0 (33.0–41.5)	36.0 (34.0–40.0)	0.740
RA size (mm) (Q_1_–Q_3_)	48.0 (42.8–53.3)	47.0 (42.0–53.0)	50.0 (46.0–58.0)	0.860
TAPSE (mm) (Q_1_–Q_3_)	15.0 (12.0–19.0)	16.1 (12.8–19.0)	13.1 (10.4–15.0)	0.026
PASP (mmHg) (Q_1_–Q_3_)	43.0 (36.0–54.0)	42.0 (35.8–53.3)	51.0 (42.5–57.5)	0.031
**Medications**				
Statins, n (%)	80 (43.7)	66 (40.4)	14 (70.0)	0.012
Antiplatelet agents, n (%)	48 (26.0)	39 (23.7)	9 (45.0)	0.041
Nitrates, n (%)	22 (12.2)	15 (8.9)	7 (35.0)	0.001

N, number of patients in the group; n, number of cases; Q_1_/Q_3_, first/third quartile; CRT-P, cardiac resynchronizating device without defibrillator; NYHA, New York Heart Association; LVEF, left ventricular ejection fraction; MR, mitral regurgitation; LVEDD, left ventricular end-diastolic diameter; LVEDDi, left ventricular end-diastolic diameter index; LVEDV, left ventricular end-diastolic volume; LVEDVi, left ventricular end-diastolic volume index; LVESV, left ventricular end-systolic volume; LVESVi, left ventricular end-systolic volume index; LA, left atrium; RV, right ventricular; RA, right atrium; TAPSE, tricuspid annular plane systolic excursion; PASP, pulmonary artery systolic pressure.

**Table 2 medicina-56-00019-t002:** Receiver operating characteristics analysis to find out diagnostic accuracy of right ventricular function parameters in predicting survival after cardiac resynchronization.

Characteristic	Cut-Off Value	Sensitivity	Specificity	The area Under the Curve (AUC)	*p*
TAPSE (mm)	15.5	0.824	0.526	0.678	0.018
PASP (mmHg)	39.5	0.909	0.519	0.733	0.013

TAPSE, tricuspid annular plane systolic excursion; PASP, pulmonary artery systolic pressure.

**Table 3 medicina-56-00019-t003:** Univariate and multivariate predictors of mortality after cardiac resynchronization.

Characteristic	Univariate Cox Regression			Multivariate Cox Regression		
	HR	95% CI	*p*	aHR	95% CI	*p*
Male gender	6.068	0.812–45.345	0.079	-	-	-
Ischemic etiology	5.134	1.496–17.625	0.009	15.235	1.999–116.088	0.009
TAPSE <15.5 mm	5.019	1.436–17.539	0.012	-	-	-
PASP ≥39.5 mmHg	2.681	0.769–9.343	0.122	-	-	-
Antiplatelet agents	2.569	1.060–6.226	0.037	-	-	-
Statins	2.983	1.146–7.764	0.025	-	-	-
Nitrates	3.694	1.342–10.171	0.011	-	-	-

HR, hazard ratio; aHR, adjusted hazard ratio; CI, confidence interval; TAPSE, tricuspid annular plane systolic excursion; PASP, pulmonary artery systolic pressure.

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
