# Peer review of "Prognostication of Poor Survival After Cardiac Resynchronization Therapy"

_medicina, 2020, doi:10.3390/medicina56010019_

Round 1

Reviewer 1 Report

The authors examined the significant prognostic factors in patients underwent CRT implantation. They found that 20 out of 183 patients died during follow-up. They found that the ischemic origin of heart failure was an independent predictor of mortality as well as TAPSE and the use of some medications. However, an increase of LVEF by the CRT was not a marker of survival. This is a retrospective and small-sized study.

Please specify the cause of death.

Were there any lost for follow up in this study? Also, it was not clear how the authors handle the missing data. Please provide relevant information.

It seems to be established that ischemic origin heart failure has an important prognostic implication. What is the strength of this study? The authors can be more explicit about the particular contribution this study can make.

Suggest combining Table 1 and Table 2. Please put the Table 1 in the leftmost column of Table 2.

Suggest adding the way of multivariate Cox analysis. How did you adjust for the multivariate analysis?

The logic of the Results does not flow very well. Suggest changing the order of Tables and Figures as follows. Table 4, Table 3, and Figure 2. The TAPSE and PASP were not independent predictor, then please delete Figures 3 and 4.

Suggest deleting univariate analysis of 3 drugs, because this data shows that they worsen the survival.

Author Response

Dear Sir or Madam,

We are much obliged to You for Your comments and remarks. We took them seriously and tried to address them as best as we can. We attach our responses to Your comments in a Word file. If we missed any points or You have further comments, we kindly consider them after Your next review.

Because we had two revisions, we had made rather more changes than You had recommended:

complemented methodology, inserted a few additional remarks about the study's limitations.

We hope, our article looks now better before.

Kind regards,

Paulius Basinskas

Reviewer 2 Report

Thanks for the opportunity to review this nice study. Please, consider the following issues:

-As a retrospective cohort study design, this manuscript should follow the STROBE criteria, which should be followed and cited.

-Study population characteristics, BMI was not included, why? This would be useful for prediction analyses.

-The procedure for the cohort needs to be clearly stated

-Nice statatistical anlyses

-In results section, there is a missing citation error ("(Error! Reference source not found.)"). In addition, please do not introduce references in results section

-Please, discuss about future studies should include BMI for prediction analyses (Eur J Heart Fail. 2019 Sep;21(9):1093-1102) and the influence of psychological variables and care during intensive care units stay after cardiac interventions (Intensive Crit Care Nurs. 2019 Oct;54:46-53).

Author Response

Dear Sir or Madam,

We are much obliged to You for Your comments and remarks. We took them seriously and tried to address them as best as we can. We attach our responses to Your comments in a Word file. If we missed any points or You have further comments, we kindly consider them after Your next review.

Because we had two revisions, we had made rather more changes than You had recommended:

we had merged Table 1 with Table 2, removed Figure 4.

We hope, our article looks now better before.

Kind regards,

Paulius Basinskas
